# Assessing reliability in neuroimaging research through intra-class effect decomposition (ICED)

Andreas M Brandmaier[1,2]*, Elisabeth Wenger[1], Nils C Bodammer[1], Simone Kühn[3], Naftali Raz[1,4], Ulman Lindenberger[1,2]

[1]Center for Lifespan Psychology, Max Planck Institute for Human Development, Berlin, Germany; [2]Max Planck UCL Centre for Computational Psychiatry and Ageing Research, Berlin, Germany; [3]Clinic and Policlinic for Psychiatry and Psychotherapy, University Clinic Hamburg-Eppendorf, Hamburg, Germany; [4]Wayne State University, Detroit, United States

**Abstract** Magnetic resonance imaging has become an indispensable tool for studying associations of structural and functional properties of the brain with behavior in humans. However, generally recognized standards for assessing and reporting the reliability of these techniques are still lacking. Here, we introduce a new approach for assessing and reporting reliability, termed intra-class effect decomposition (ICED). ICED uses structural equation modeling of data from a repeated-measures design to decompose reliability into orthogonal sources of measurement error that are associated with different characteristics of the measurements, for example, session, day, or scanning site. This allows researchers to describe the magnitude of different error components, make inferences about error sources, and inform them in planning future studies. We apply ICED to published measurements of myelin content and resting state functional connectivity. These examples illustrate how longitudinal data can be leveraged separately or conjointly with cross-sectional data to obtain more precise estimates of reliability.

DOI: https://doi.org/10.7554/eLife.35718.001

*For correspondence:
brandmaier@mpib-berlin.mpg.de

**Competing interests:** The authors declare that no competing interests exist.

## Introduction

Neuroimaging techniques have become indispensable tools for studying associations among brain structure, brain function, and behavior in multiple contexts, including aging, child development, neuropathology and interventions, with concerted efforts increasingly focusing on comprehensive quantitative analyses across multiple imaging modalities (*Lerch et al., 2017*). Surprisingly, however, generally recognized standards and procedures for assessing and reporting the reliability of measurements and indices generated by noninvasive neuroimaging techniques are still lacking. This state of affairs may reflect the rapid evolution of a research field that straddles several well-established disciplines such as physics, biology, and psychology. Each of these fields comes with its own methodology, including conceptualization of error of measurement and reliability, and an articulation of these diverse methodologies into a coherent neuroscience framework is currently lacking. The goal of our contribution is two-fold. First, we introduce a signal-to-noise perspective that reconciles these seemingly disparate approaches. Second, we apply an analytic framework, based on the ideas of Generalizability Theory (G-Theory; *Cronbach et al., 1972*) and Structural Equation Modeling (SEM) that allows us to separate and gauge various sources of measurement error associated with different characteristics of the measurement, such as run, session, day, or scanning site (in multi-site studies). The proposed tool enables researchers to describe the magnitude of individual error components, make inferences about the error sources, and inform them in planning the design of future studies.

We proceed without loss of generality but with an emphasis on applications to human cognitive neuroscience.

## Materials and methods

### Prelude: Coefficient of variation and intra-class correlation coefficient represent different but compatible conceptions of signal and noise

Physics and psychometrics offer two fundamentally different but equally important and compatible conceptions of reliability and error. Physicists typically inquire how reliably a given measurement instrument can detect a given quantity. To this end, they repeatedly measure a property of an object, be it a phantom or a single research participant, and for expressing the absolute precision of measurement, evaluate the dispersion of the different measurement values obtained from this object to their mean. The prototypical index produced by such approach is the coefficient of variation (CV), which is defined as the ratio of an estimate of variability, $\sigma_i$, and a mean, $m$, with $i$ representing the object undergoing repeated measurements:

$$CV_i = \frac{\sigma_i}{m_i}$$

The interpretability of the CV depends upon the quantity having positive values and being measured on a ratio scale. When these conditions are met, the CV effectively expresses the (im)precision of measurement, with larger values meaning lesser precision. Imagine, for instance, that the same quantity is being measured in the same research participant or the same phantom on two different scanners. All other things equal, comparing the CV obtained from each of the two scanners shows which of the two provides a more reliable (in this case, precise) measurement.

Note that in this context, the scanner with the greater precision may not necessarily yield more valid data, as the mean of its measurements may be further away from the ground truth (see *Figure 1*). Bearing this distinction in mind, we limit our discussion to the issues of reliability (precision), rather than validity (bias). In *Table 1*, we list terms used in various disciplines to express the difference between precision and bias. We maintain that the confusion surrounding these concepts may to a large extent reflect terminological differences among disciplines.

In contrast to physics that deals with well-defined objects of measurement, in human neuroscience, we focus on a different meaning of reliability. Informed by psychometric theory and differential psychology, reliability here refers to the precision of assessing between-person differences. Researchers concerned with gauging individual differences as a meaningful objective express this form of reliability in a ratio index, termed intra-class correlation coefficient (ICC), which represents the strength of association between any pair of measurements made on the same object. However, instead of relating variance to the mean, the ICC quantifies variance *within* persons (or groups of persons), in relation to the total variance, which also contains variance *between* persons (or between groups of persons; cf. *Bartko, 1966*). Hence, the ICC is a dimensionless quantity bracketed between 0 and 1, and is tantamount to the ratio of variance-between, $\sigma_B^2$, to the total variance that includes the variance-within, $\sigma_W^2$:

$$ICC = \frac{\sigma_B^2}{\sigma_W^2 + \sigma_B^2}$$

In repeated-measures studies on human participants, the variance-within corresponds to the variance within each person, whereas the variance-between represents differences among persons. Thus, for interval or ratio scales, the ICC expresses the percentage of the total variance that can be attributed to differences between persons.

The similarities and differences between CV and ICC become clear when one conceives of both as expressions of signal-to-noise ratio. For a physicist, the mean represents the sought-after signal, and the variation around the mean represents the noise to be minimized. Hence the use of the CV to evaluate measurement precision normalized with the metric of the given scale. For a psychologist interested in individual differences, the between-person variation is the signal, and the within-person variation is regarded as noise. Therefore, a measure that quantifies the contribution of between-person differences to the total variance in the data, the ICC, is chosen for this purpose (in other

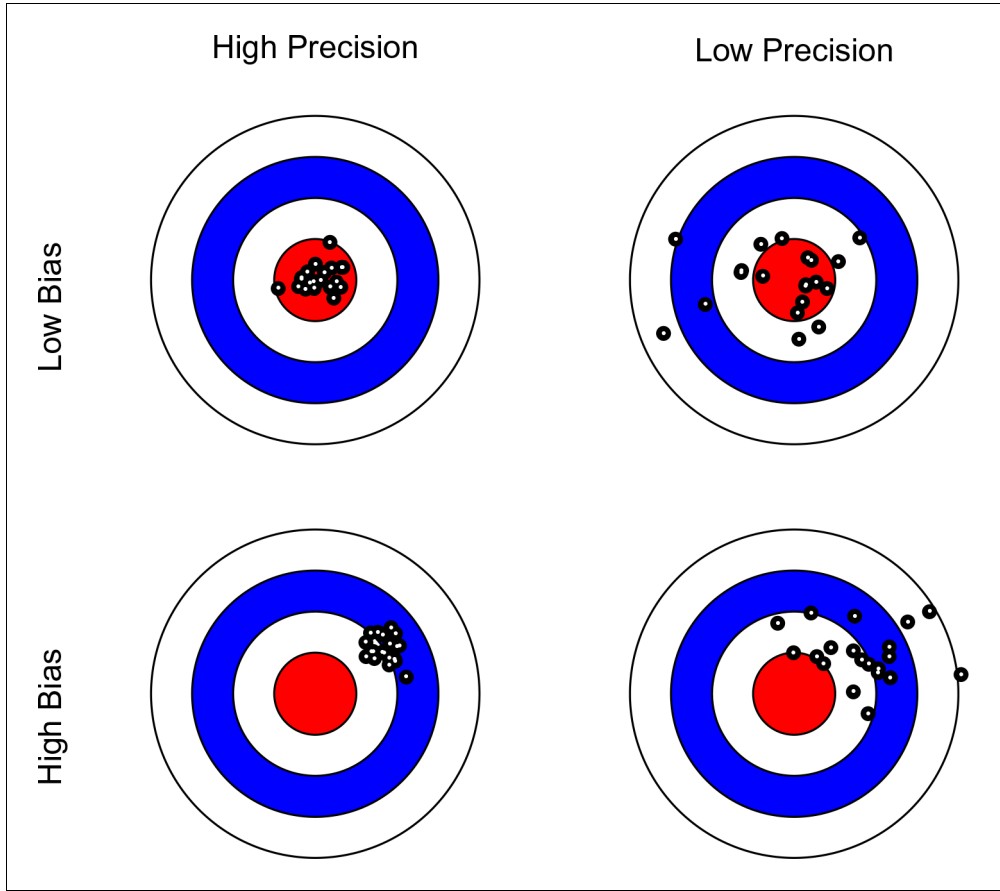

**Figure 1.** Bullseye charts representing precision and bias of a measurement instrument. The center of each bullseye represents ground truth and the black dots represent repeated measurements. Ideal measurement instruments have minimal bias and maximal precision (as illustrated in the top-left panel).

DOI: https://doi.org/10.7554/eLife.35718.002

contexts, not discussed in this article, within-person variability itself may be an important marker of individual differences, e.g., *Garrett et al., 2013*; *Nesselroade, 1991*).

Clearly, CV and ICC do not convey the same information. To illustrate this point, we simulated data under two conditions, which show that each measure can be manipulated independently of the other. We illustrate how CV remains unchanged, while drastic changes occur in ICC (see *Figure 2*). Instead of individual CV values, we report an aggregated CV computed as the square-root of the average within-person variance divided by the overall mean. For each condition, we simulated for each of five persons ten repeated measures of a fictitious continuous outcome variable $X$. Across conditions and persons, within-person variability was identical and only between-person variability

**Table 1.** A list of terms describing the concepts of variability across repetitions and average distance from ground truth over measurements across different disciplines and knowledge domains.

| Knowledge domain | Variability across repetitions | Average distance from ground truth |
| --- | --- | --- |
| Psychology | Reliability | Validity |
| Physics | Precision | Accuracy |
| Statistics | Variance | Bias |
| Measurement Theory | Random error | Systematic error |
| ISO 5725 ('Accuracy') | Precision | Trueness |

DOI: https://doi.org/10.7554/eLife.35718.003

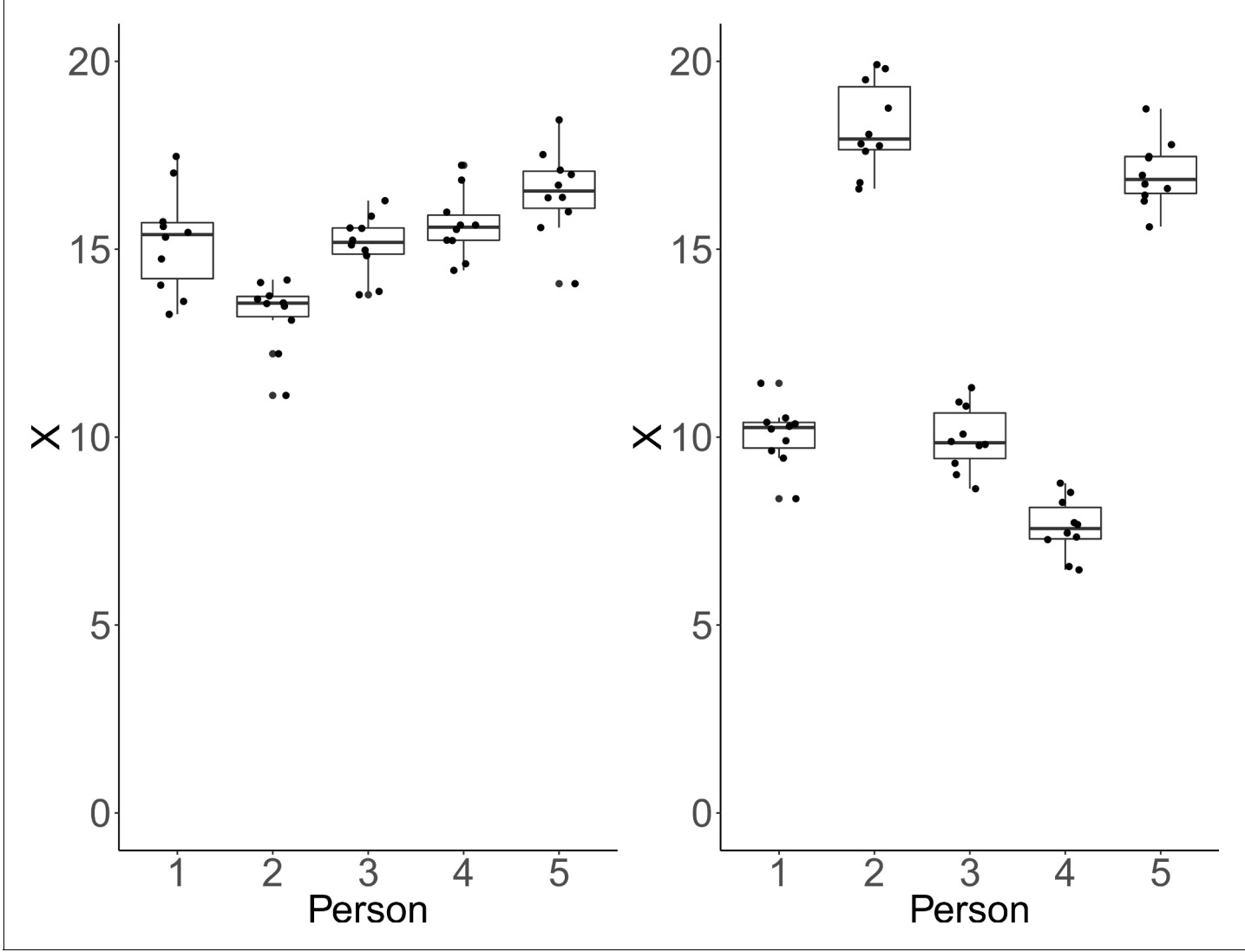

**Figure 2.** Each plot shows simulated data of 10 repeated measurements of a fictitious outcome X for five persons. Within-person variance is identical over persons and panels. Left panel: Between-person variance is identical to within-person variance. In that case, ICC-indexed reliability is estimated at a rather low ICC = 0.48, whereas CV-indexed precision is fair, with averaged CV = 0.07. Right panel: Between-person standard deviation is larger than within-person standard deviation by a factor of five. In this condition, estimated ICC-indexed reliability is high (ICC = 0.96), but CV-indexed precision remains identical with averaged CV = 0.07 as the within-person standard deviation is still fairly small with respect to the sample mean.
DOI: https://doi.org/10.7554/eLife.35718.004

varied between conditions. In the first condition, the simulated data have identical between-person and within-person variance. As a result, we obtain a low ICC and conclude that the measurement instrument fails to adequately discriminate among persons. However, critically, we also obtain a rather low CV, implying high precision to detect deviation from zero (see left panel of *Figure 2*). In the second instance, between-person standard deviation was larger than within-person standard deviation by a factor of five. This condition yields a high ICC reflecting the fact that the measure discriminates well among persons. At the same time, CV remains low, which implies reasonable precision of detecting differences from zero. This is because the within-person variance is still relatively low in comparison to the means (see right panel of *Figure 2*).

In summary, whereas the CV refers to the precision of measurement obtained from each object, the ICC expresses a part-whole relation of variance observed in the data. All other things being equal, a less precise measurement will increase the variance-within, and hence compromise our ability to detect between-person differences. On the other hand, a rather imprecise measurement (as

indexed by the CV obtained for each object of measurement) may nevertheless yield high reliability (as indexed by the ICC) if the between-person differences in means are large.

## Intra-class effect decomposition (ICED)

The extant neuroimaging literature typically offers little justification for the choice of the reliability index. Based on the preceding considerations, this is problematic, as the various indices differ greatly in meaning. The ICC and variants thereof are appropriate for evaluating how well one can capture between-person differences in a measure of interest. Put differently, it is misleading to report the CV as a measure of reliability when the goal of the research is to investigate individual or group differences. Both approaches to reliability assessment are informative, but they serve different purposes, and cannot be used interchangeably. Below, we focus on individual differences as we present a general and versatile method for estimating the relative contributions of different sources of error in repeated-measures designs. Our approach can be seen as an extension of ANOVA-based approaches to decomposing ICCs. In this sense, it is tightly linked to G-Theory, which has been used successfully before in assessing reliability of neuroimaging measures (*Gee et al., 2015*; *Noble et al., 2017*). The method, termed intra-class effect decomposition (ICED), has ICC as its core concept. The key feature of the method, however, is its ability to distinguish among *multiple* sources of un-reliability, with the understanding that not all sources of error and their separation are important and meaningful in repeated-measures designs. For example, different sources of error may be due to run, session, day, site, scanner, or acquisition protocol variations. Furthermore, there may be more complex error structures to be accounted for, for example, runs nested in sessions; and multiple sessions, again, may be nested within days, and all may be nested under specific scanners in multi-site investigations. Neglecting these nuances of error structures leads to biased reliability estimates. The ability to adequately model these relationships and visually represent them in path diagrams is a virtue of our approach.

Beyond reliability per se, researchers may often be interested in the specific sources of error variance and measurement characteristics that contribute to it. For example, in applying MRI to studying long-term within-person changes in the course of aging, child development, disease progression, or treatment, one may wish to determine first what effect repositioning of a person in the scanner between sessions has on reliability of measured quantities (e.g., *Arshad et al., 2017*). Similarly, it may be important to determine how much variation is associated with scanning on a different day relative to conducting two scanning sessions on the same day (e.g., *Morey et al., 2010*). These types of questions are of utmost importance in longitudinal studies, in which researchers collect data on the same person using an ostensibly identical instrument (e.g., MRI scanner ) under an identical protocol (sequence), but inevitably under slightly different measurement characteristics, including position of the participant within the scanner, body and air temperature, or time of day. From a design perspective, knowing the distinct components of measurement error and their relative magnitudes may enhance future study designs and boost their generalizability.

In the proposed SEM framework, observed variance is partitioned into several orthogonal error variance components that capture unreliability attributable to specific measurement characteristics, with the number of components depending on identification constraints based on the study design. *Figure 3* shows a minimal, or optimally efficient, repeated-measures study design for estimating the contributions of the main effects of day, session, and residual variance to measurement error. The design consists of four measurements (scans) performed over two days and three sessions. In this design, unique contributions of each error source are identified as depicted in the path diagram in *Figure 4*. In the diagram, observed variables correspond to image acquisitions and are depicted as rectangles; latent variables are depicted as circles and represent the unobservable sources of variance, that is, the true score variance (T) and the error variance components of day (D), session (S), and residual (E). Double-headed arrows represent variances of a latent variable. Single-headed arrows represent regressions with fixed unit loadings.

In this example, total observed variability in an outcome across measurements and persons is partitioned into true-score variance and three error variance terms: the day-specific error variance, the session-specific error variance (here capturing the effect of repositioning a person between scans), and the residual error variance. The full measurement model is depicted as a path diagram in the left panel of *Figure 4*. The structural equation model specifies four observed variables representing the repeated measurements of the outcome of interest. One of the latent variables represents the

**Figure 3.** A study design with four brain imaging scans per person spread across three sessions on two days. The start of each session (blue) implies that the person is moved into the scanner. On the first day, there is only a single session, that is, between scans 1 and 2, the person remains in the scanner whereas on day 2, the person is removed from the scanner after the first session and, after a short break, placed back in. This allows separating the session-specific and the day-specific variance contributions to total variance.
DOI: https://doi.org/10.7554/eLife.35718.005

true values of the construct of interest. Its variance, $\sigma_T^2$, denotes the between-person variance. Fixed regressions of each measurement occasion on the latent construct express the assumption that we are measuring the given construct with each of the four repeated measures on the same scale. There are four orthogonal error variance sources with identical residual variance $\sigma_E^2$, that is, residual errors that are not correlated with any other type of error or among themselves over time. In classical test theory, this is referred to as a *parallel model*, in which the construct is measured on the same scale with identical precision at each occasion. Typically, there is no explicit assumption of uncorrelated error terms even though many measures derived from this theory assume (and are only valid under) uncorrelated error terms (*Raykov et al., 2015*). Here, we focus on a parallel model while accounting for the correlated error structure implied by the greater similarity of multiple runs within the same session compared to runs across different sessions. Note that, in the SEM framework, we also can extend the parallel model to more complex types of measurement models (e.g., con-generic or tau-equivalent models) that allow for different residual error variances or different factor loadings. To account for the nested structure in our design, we introduce two day-specific error variance sources with variance $\sigma_D^2$ that represent day-specific disturbances and imply a closer similarity of measurements on the same day. Finally, there are three session-specific variance sources (depicted in blue) representing the session effect (including, for example, the effect of repositioning a person between sessions). The model-implied covariance matrix has the total variances for each observed variable in the diagonal. It can be analytically or numerically derived using matrix algebra (*McArdle, 1980*) or path-tracing rules (*Boker et al., 2002*), and is typically available in SEM computer programs (e.g., *von Oertzen et al., 2015*). The full model-implied covariance matrix is given in *Table 2*. For the given study design, each variance source is uniquely identifiable, as there is a unique solution for all parameters in the model. From the covariance matrix, it is apparent that the inclusion of the session variance term differentially affects the similarity of measurements between days 1 and 2. The

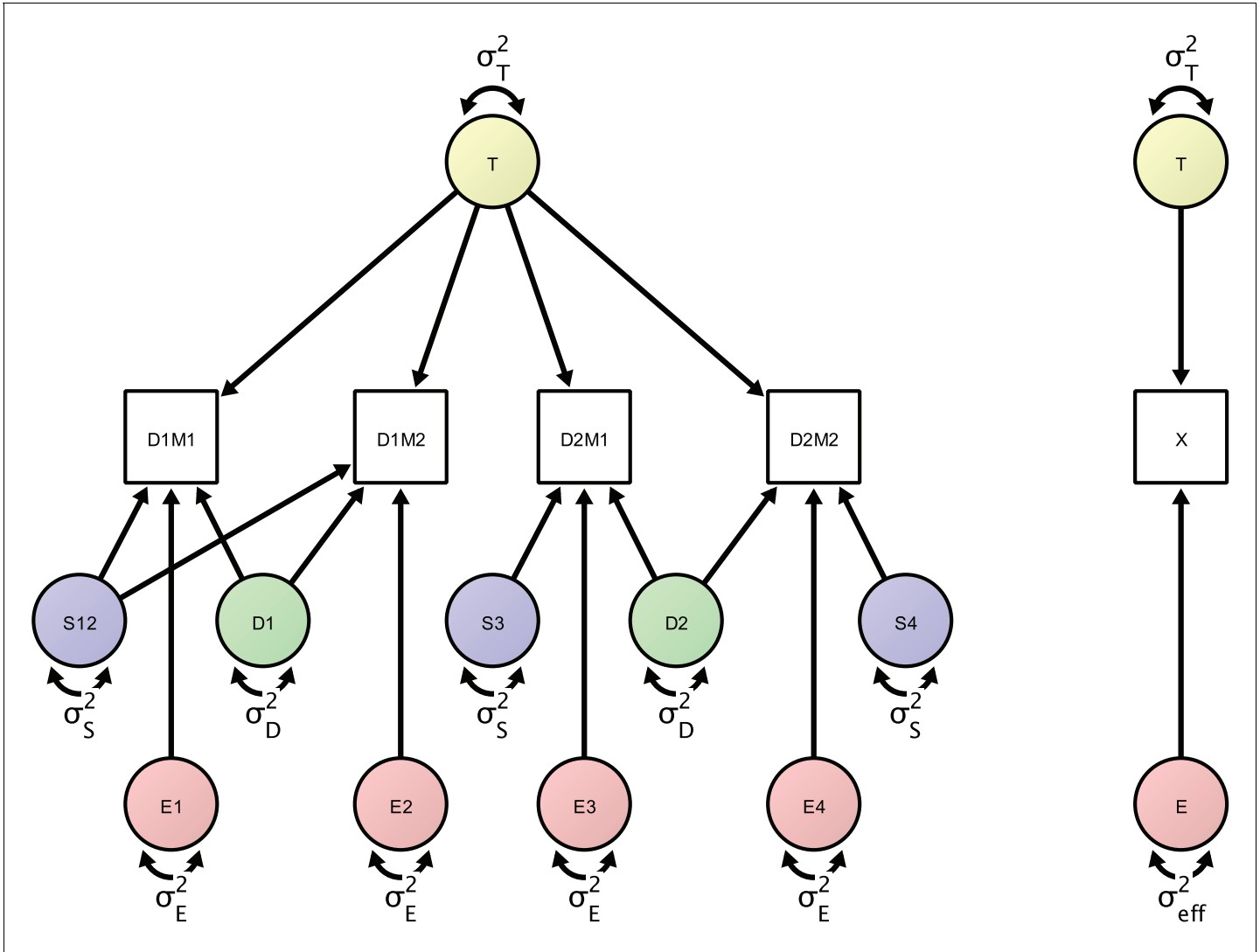

**Figure 4.** Left: Measurement model of four repeated measures according to the study design protocol. The measurement model includes day-specific effects that are separated by the error variance sources (in green), session-specific effects (represented by error variance sources in blue) and orthogonal residual error variance sources (represented in red). Right: A minimal, power-equivalent model that represents a direct measurement of the construct of interest. The true score (representing the outcome of interest) is measured with only a single error construct, whose variance is the effective error variance representing the combined influence of the complete error (co)variance structure shown in the left model.

DOI: https://doi.org/10.7554/eLife.35718.006

correlation between first and second scan is $\frac{\sigma_T^2+\sigma_D^2+\sigma_S^2}{\sigma_T^2+\sigma_E^2+\sigma_D^2+\sigma_S^2}$ whereas the correlation between measurements 3 and 4 is $\frac{\sigma_T^2+\sigma_D^2}{\sigma_T^2+\sigma_E^2+\sigma_D^2+\sigma_S^2}$. Thus the similarity of the two measurements on the first day is greater than the similarity of measurements on the second day. In other words, the difference in correlation is the proportion of variance that the session-specific variance accounts for in total variance.

For this model (see **Figure 4**), we define ICC equivalently to the common ICC formula as ratio of between-person variance to total variance at the level of observed variables:

$$ICC = \frac{\sigma_T^2}{\sigma_T^2 + \sigma_E^2 + \sigma_D^2 + \sigma_S^2}$$

We estimate the components using the full information maximum likelihood procedure for SEM (**Finkbeiner, 1979**), which allows estimating all components under the assumption of the data

**Table 2** Model-implied covariance matrix.

Rows and columns correspond to the four measurement occasions (brain scans) distributed over two days. Parameters of the covariance matrix are the individual differences of the construct of interest, $\sigma_T^2$, the session-specific error variance, $\sigma_S^2$, the day-specific error variance, $\sigma_D^2$, and the residual error variance, $\sigma_E^2$.

|  | Scan 1 | Scan 2 | Scan 3 | Scan 4 |
|---|---|---|---|---|
| Scan 1 | $\sigma_T^2 + \sigma_E^2 + \sigma_D^2 + \sigma_S^2$ | $\sigma_T^2 + \sigma_D^2 + \sigma_S^2$ | $\sigma_T^2$ | $\sigma_T^2$ |
| Scan 2 | $\sigma_T^2 + \sigma_D^2 + \sigma_S^2$ | $\sigma_T^2 + \sigma_E^2 + \sigma_D^2 + \sigma_S^2$ | $\sigma_T^2$ | $\sigma_T^2$ |
| Scan 3 | $\sigma_T^2$ | $\sigma_T^2$ | $\sigma_T^2 + \sigma_E^2 + \sigma_D^2 + \sigma_S^2$ | $\sigma_T^2 + \sigma_D^2$ |
| Scan 4 | $\sigma_T^2$ | $\sigma_T^2$ | $\sigma_T^2 + \sigma_D^2$ | $\sigma_T^2 + \sigma_E^2 + \sigma_D^2 + \sigma_S^2$ |

DOI: https://doi.org/10.7554/eLife.35718.007

missing at random. This maximum-likelihood-based ICC is similar to the analytical procedure based on relating the ANOVA-derived within and between residual-sums-of-squares. The main difference is that the maximum likelihood estimator cannot attain negative values when we allow only positive variance estimates (*Pannunzi et al., 2018*).

In many cognitive neuroscience studies, one may be interested in construct-level reliability, and not only in reliability of indicators (i.e., observed variables). This construct reliability is captured by $ICC_2$ (*Bliese, 2000*). Based on the above SEM-based effect decomposition, we use power equivalence theory (*von Oertzen, 2010*) to derive the *effective error* of measuring the latent construct of interest. The effective error can be regarded as the residual error that would emerge from a direct measurement of a latent construct of interest. Here, it is an index of the precision with which a given study design is able to capture stable individual differences in the outcome of interest. The effective error is a function of all error components and its specific composition depends on the specific design in question. Effective error is the single residual error term that arises from all variances components other than the construct that is to be measured. As such, it represents the combined influence of all error variance components that determine construct reliability:

$$ICC_2 = \frac{\sigma_T^2}{\sigma_T^2 + \sigma_{eff}^2}$$

Effective error can be computed using the algorithm provided by *von Oertzen (2010)* and for some models, analytic expressions are available (see the multi-indicator theorem in *von Oertzen, 2010*). For the study design in our example, effective error is:

$$\sigma_{eff}^2 = \frac{\left(2\sigma_D^2 + \sigma_E^2 + \sigma_S^2\right)\left(2\sigma_D^2 + \sigma_E^2 + 2\sigma_S^2\right)}{8\sigma_D^2 + 4\sigma_E^2 + 3\sigma_S^2}$$

Relating true score variance to total variance yields $ICC_2$ – a measure of reliability on the construct level. For our model, $ICC_2$ is then:

$$ICC_2 = \frac{\sigma_T^2}{\sigma_T^2 + \frac{\left(2\sigma_D^2 + \sigma_E^2 + \sigma_S^2\right)\left(2\sigma_D^2 + \sigma_E^2 + 2\sigma_S^2\right)}{\left(8\sigma_D^2 + 4\sigma_E^2 + 3\sigma_S^2\right)}}$$

As a check, when assuming no day-specific and session-specific effects by inserting $\sigma_D^2 = 0$ and $\sigma_S^2 = 0$, we obtain the classical definition of $ICC_2$ that scales residual error variance with the number of measurement occasions (here, four occasions):

$$ICC_2 = \frac{\sigma_T^2}{\sigma_T^2 + \frac{\sigma_E^2}{4}}$$

In sum, ICC is a coefficient describing test-retest reliability of a measure (also referred to as short-term reliability or intra-session reliability by *Noble et al., 2017*) whereas $ICC_2$ is a coefficient

describing test-retest reliability of an underlying construct (an average score in parallel models) in a repeated-measures design (long-term reliability or intersession reliability according to *Noble et al., 2017*).

For our hypothesized measurement model that includes multiple measurements and multiple variance sources, the analytic solution of $ICC_2$ allows, for instance, to analytically trace reliability curves depending on properties of a design, such as the number of sessions, number of runs per sessions, number of sessions per day, or varying magnitudes of the error component. Of note, this corresponds to a D-study in G-Theory that can demonstrate, for example, how total session duration and number of sessions influence resting state functional connectivity reliability (see *Noble et al., 2017*).

A virtue of the proposed SEM approach is the possibility of applying likelihood-ratio tests to efficiently test simple and complex hypotheses about the design. For example, we can assess whether individual variance components significantly differ from zero or from particular values, or whether variance components have identical contributions (corresponding to F-tests on variance components in classical G-Theory). Such likelihood-ratio tests represent statistical model comparisons between a full model, in which each of the hypothesized error components are freely estimated from the data, and a restricted model, in which the variance of a target error component is set to zero. Both models are nested, and under the null hypothesis, the difference in negative-two log-likelihoods of the models will be $\chi^2$-distributed with 1 degree of freedom. This allows the derivation of $p$ values for the null hypotheses of each individual error component being zero. Moreover, the generality of SEM allows testing complex hypotheses with hierarchically nested error structures or multi-group models. It also allows inference under missing data or by evaluating informative hypotheses (*de Schoot et al., 2011*) whereas ANOVA-based approaches become progressively invalid with increasing design complexity.

## Results

### An empirical example: Myelin water fraction data from *Arshad et al. (2017)*

To demonstrate how the proposed approach separates and quantifies sources of un-reliability, we re-analyzed data from a study of the brain regional myelin content by *Arshad et al. (2017)*. In human aging, changes of myelin structure and quantity have been proposed as neuroanatomical substrates of cognitive decline, which makes it particularly interesting to obtain a highly reliable estimate of regional myelin content, here, represented by myelin water fraction (MWF) derived from multi-component $T_2$ relaxation curves. The data in this demonstration were collected in 20 healthy adults (mean age ± SD = 45.9 ± 17.1 years, range of 24.4–69.5 years; no significant difference between men and women: t(18)=–0.81, p=0.43) and are freely available (*Arshad et al., 2018*); for detailed sample description see *Arshad et al. (2017)*. The study protocol stipulated three acquisitions for each participant in a single session. In the first part, $T_1$-weighted and $T_2$-weighted MRI images were acquired, followed by a back-to-back acquisition of the ME-$T_2$ relaxation images without repositioning the participant in the scanner. At the end of the first part, participants were removed from the scanner and, after a short break, placed back in. In the second part, $T_1$-weighted, $T_2$-weighted and ME-$T_2$ multiecho sequences were acquired once. All further details relating to the study design, MR acquisition protocol, and preprocessing can be found in the original publication by *Arshad et al. (2017)*. In the following, we focus on the MWF derived from a multi-echo gradient recall and spin-echo (GRASE) sequence. The study design allows separating the influences of repositioning expressed as session-specific variance from true score variance (defined as the shared variance over all three repetitions) and individual error variance (the orthogonal residual error structure). *Figure 5* presents a diagram of the hypothesized contributions of the individual variance components. Parameters in the SEM correspond to estimates of true score variation (*T*), session-specific error variance component (*S*), and a residual error variance component (*E*). Model specification and estimation was both performed in Ωnyx (*von Oertzen et al., 2015*) and lavaan (*Rosseel, 2012*) via full information maximum likelihood. We provide the Ωnyx models and lavaan syntax in the Supplementary material.

For illustration, we only report estimates of the first of the six regions of interest reported in the original study, the anterior limb of the internal capsule (ALIC). The estimates of the individual variance components explaining the observed variance are shown in the diagram in *Figure 5*. To assess

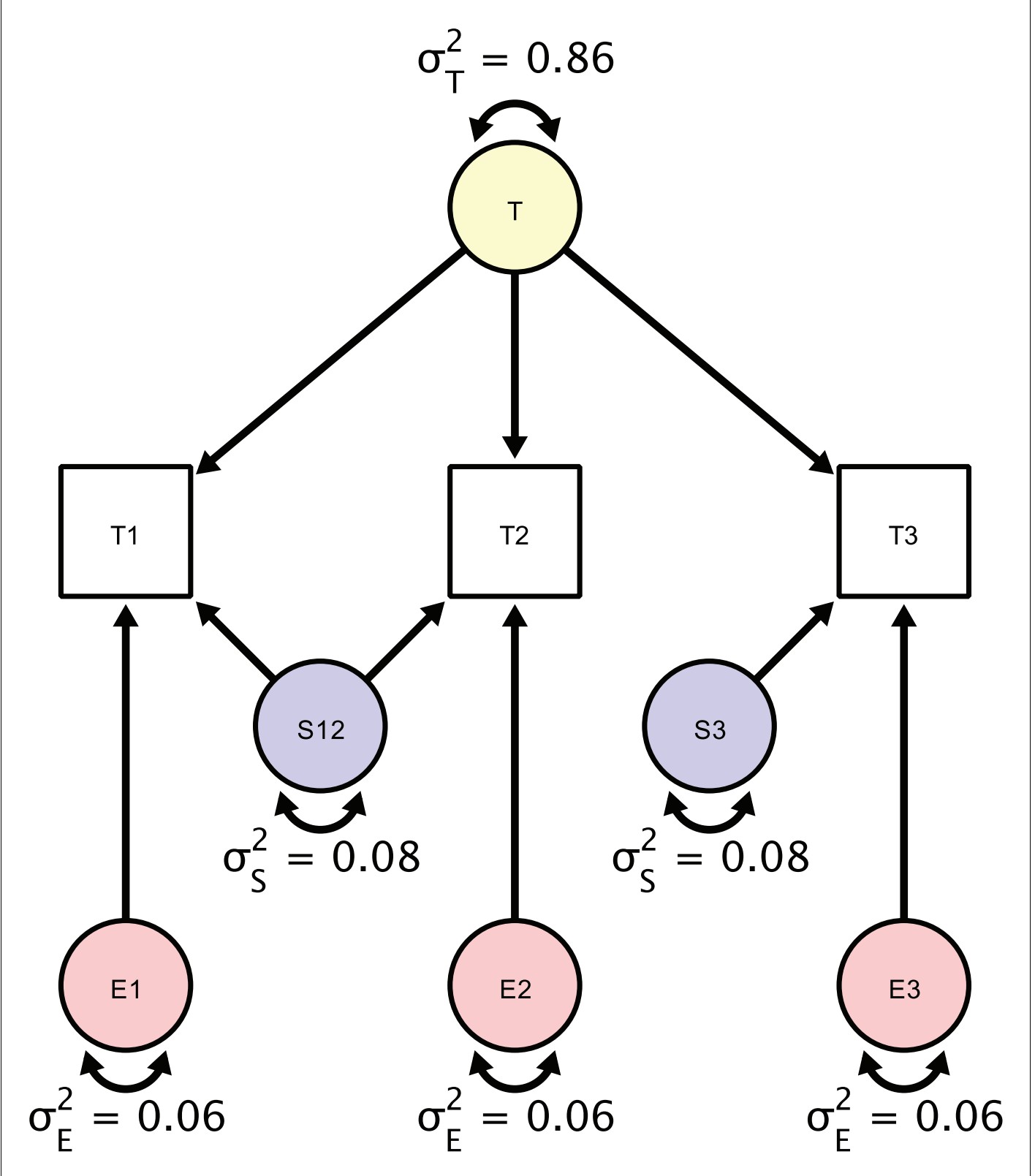

**Figure 5.** A Structural Equation Model of a repeated measures design in which each participant is scanned three times. Each person is scanned the first time (T1), followed by a back-to-back immediate re-acquisition (T2), and, finally, the person is moved out of the scanner, positioned back in the scanner, and scanned the third time (T3). Parameters in the SEM correspond to estimates of true score variation ($\sigma_T^2$), session-specific error variance

*Figure 5 continued on next page*

*Figure 5 continued*
component ($\sigma_S^2$), and a residual error variance component ($\sigma_E^2$). The standardized estimates are based on myelin water fraction data in the anterior limb of the internal capsule acquired from **Arshad et al. (2017)**.
DOI: https://doi.org/10.7554/eLife.35718.008

the significance of these components' magnitudes, we used likelihood ratio tests against null models, in which each component's variance was set to zero. For testing the residual error variance component, we used a Wald test because the null model without an orthogonal error structure cannot be estimated. Of the total between-person variance in measurements of MWF, we found that 86% were due to true score variance (est = 6.97; $\chi^2$ = 27.759; df = 1; p<0.001), 8% - to session-specific variance (est = 0.59, $\chi^2$ = 3.951; df = 1; p=0.047), and 6% - to residual error variance (est = 0.52; Z = 9.64; p=0.002). Testing whether the variance contribution of the session-specific variance and the residual error variance were equal yields a non-significant result ($\chi^2$ = 20.02; df = 1; p=0.89) and, thus, cannot be decided.

As shown before, we can obtain ICC as the ratio of systematic (*t*) and all variance components, which by means of standardization of the observed variables sums up to unity, resulting in:

$$ICC = 0.86$$

To compute ICC2 as a standardized estimate of the precision with which the repeated-measures study design can measure individual differences in MWF in ALIC, we equate the day-specific variance with zero since it is not identified in this design but rather subsumed under the estimate of the true-score variance component, yielding:

$$ICC_2 = \frac{\sigma_T^2}{\sigma_T^2 + \frac{(\sigma_E^2 + \sigma_S^2)(\sigma_E^2 + 2\sigma_S^2)}{(3\sigma_E^2 + 4\sigma_S^2)}} = 0.94$$

The fact that day-specific variance and true score variance are inseparable in this design (both are shared variance components of all three measurement occasions) leads to an inflation of the true score variance estimate if non-zero day-specific variance is assumed and, thus, to an overly optimistic estimate of reliability. To be able to separate the individual variance contributions, one would have to rely on an augmented design that includes additional scanner acquisitions on at least one different day, such as the design shown in **Figure 4**.

**Arshad et al. (2017)** only reported pair-wise ICCs, based either only on the two back-to-back sessions of a single day, or on a single session of each day (again omitting a third of the available data). In the following, we derive the corresponding pair-wise ICCs using the full data set. Our estimates are similar even though not identical to the results obtained by **Arshad et al. (2017)** because our results were jointly estimated from three measurements. First, the authors report an estimate of ICC based on one measurement from the second session of the first day and one measurement from the single session of the second day, resulting in ICC = 0.83, which is close to our estimate of ICC = 0.86. Second, they reported an estimated reliability (ICC) derived only from the two back-to-back sessions on the first day as ICC = 0.94, Similarly, we can derive the reliability of a single measurement, had we measured only the two back-to-back sessions, achieving the identical result:

$$ICC = \frac{\sigma_T^2 + \sigma_S^2}{\sigma_T^2 + \sigma_S^2 + \sigma_E^2} = 0.94$$

The estimates of construct-level reliability obtained imply that individual differences in MWF can be measured quite well. As expected, the reliability estimate is higher for the back-to-back session than for the complete design because one error variance component, session-specific error variance, is not apportioned to the total error variance. Such a simple design commingles true score variance and the session-specific variance, and reliability studies should thus, by design, take into account potential different error sources, such as session-specific error variance.

A comprehensive SEM approach to assessing reliability allows for using the complete dataset in a single model to estimate reliability as either item-level reliability (ICC) or a construct-level reliability (ICC$_2$). A particular benefit of the proposed approach is its ability to tease apart individual error

components as far as the study design permits this, that is, as far as these components are identi-fied. Future studies may very well increase study design complexity to test for additional error vari-ance components. To compare the effect of repositioning a participant versus scanning a participant back-to-back, *Arshad et al. (2017)* compared pairwise ICCs of either the two back-to-back acquisi-tions or the second of the back-to-back acquisition with the repositioned acquisition. Using the SEM-based approach described above, we can directly estimate a variance component that quanti-fies the contribution of the session to the total error variance. We can also formally test whether this contribution is non-zero, or, if necessary, whether it is greater than some value or some other error variance component in the model. Furthermore, our estimates are always based on the complete dataset and there is no need to select certain pairs of runs for computing subset ICCs and poten-tially disregarding important dependencies in the data – a limitation that *Arshad et al. (2017)* explicitly mentioned in their report.

## Link-wise reliability of resting state functional-connectivity indices

Resting-state functional connectivity was proposed as a promising index of age-related or pathol-ogy-induced changes in the brain, and has been used to predict brain maturation (*Dosenbach et al., 2010*) or disease state (*Craddock et al., 2009*). These applications can only prove practically useful if reliability is sufficiently high, so that differences between persons can be reliably detected in the first place, as a methodological precondition for prediction. Thus, there has been increasing interest in examining reliability of methods for assessing resting state connectivity (*Gordon et al., 2017*; *Noble et al., 2017*; *Pannunzi et al., 2018*). Here, we demonstrate how ICED can be used to evalu-ate reliability of pairwise functional indices obtained from resting-state functional connectivity analyses.

To illustrate such a model, we obtained the resting state functional connectivity (rsFC) dataset from *Pannunzi et al., 2018*, which is based on the publicly available raw data from the Day2day study (*Filevich et al., 2017*). In that study, six participants were scanned at least 40 (and some up to 50) times over the course of approximately seven months, and another sample of 50 participants (data from 42 participants of them available) were each scanned only once. In the following, we show how both datasets can be jointly investigated to estimate link-wise reliability of resting state functional connectivity (rsFC). We present a reliability analysis of the link-wise connectivity indices of brain regions-of-interest based on 5 min of measurement. For each measurement, as our main out-come, we obtained a $16 \times 16$ correlation matrix of rsFC indices, for pairs of regions including pre-frontal, sensor-motor, parietal, temporal, limbic, occipital cortices, cerebellum and subcortical struc-tures. In our model, we assume independence of the measurement occasions. Thus, we decompose the covariance structure of the repeated measurements into one between-person variance and one within-person variance component. For simplicity, we illustrate this model by using the first ten observations. *Figure 6* shows a path diagram of this model. We estimated this model using Ωnyx and lavaan and significance tests were performed using Wald tests. For example, we first estimated our model only for the link between left prefrontal cortex and right prefrontal cortex. The true score variance was estimated to account for 49% of the total variance (est = 0.013; W = 2.46; df = 1; p=0.117) and the error variance contributed 51% of the total variance (est = 0.014; W = 27.00; df = 1; p<0.0001), thus, ICC was 0.49.

With up to fifty measurement occasions, we can expect to get sufficiently precise measures of within-person fluctuations but since only eight participants contributed, we augment this dataset with cross-sectional data from additional 42 persons treating them as quasi-longitudinal data with the majority of data missing. This more precise measurement of between-person differences yields a somewhat different pattern of results. The true score variance was 39% of the total variance (est = 0.008; W = 6.31; df = 1; p=0.012) and the error variance was 61% of the total variance (est = 0.013; W = 33.53; df = 1; p<0.0001). Thus, our estimate dropped from 0.49 to 0.39. Due to a small sample size in the first analysis, we likely had overestimated the between-person differences in rsFC and had obtained an exceedingly overoptimistic ICC. By augmenting the initial analysis with a second dataset, we have obtained more precise and, here, even more pessimistic estimates of rsFC reliability.

*Figure 7* shows a reliability matrix of all links between the investigated brain regions with esti-mates based on the joint model. *Pannunzi et al., 2018* reported that ICCs range from 0.0 to 0.7 with an average ICC of 0.22, which is typically considered an unacceptably low reliability (i.e., signal

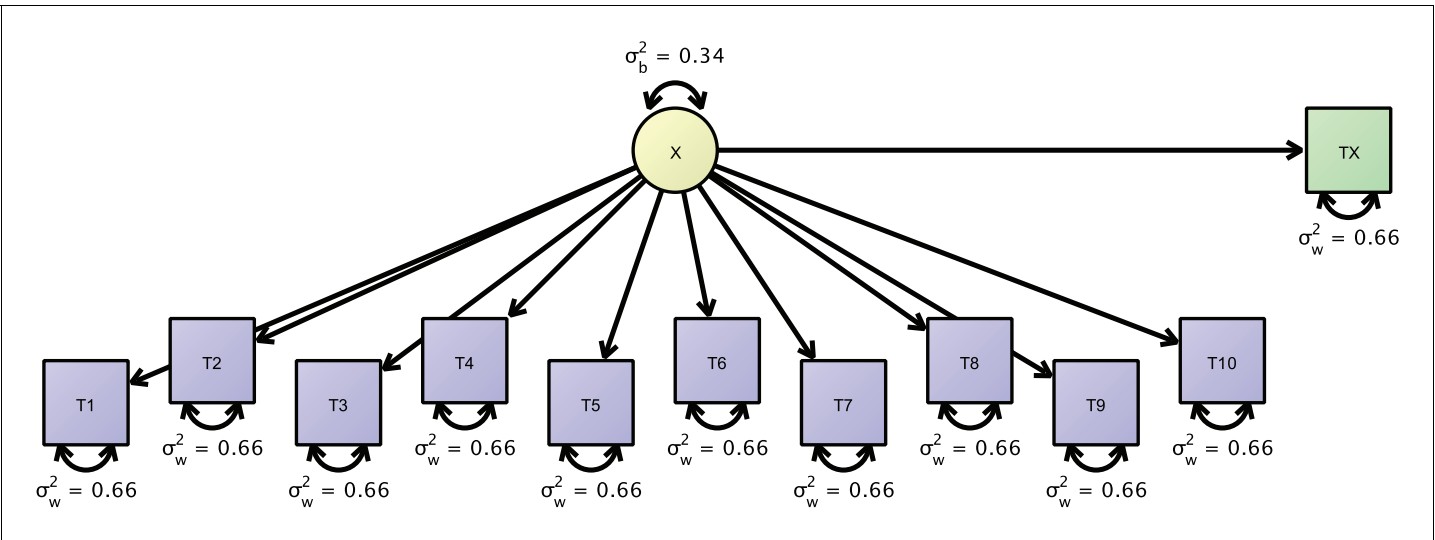

**Figure 6.** Path-diagram of a joint cross-sectional and longitudinal model to estimate link-wise resting state functional connectivity. The latent variable X (yellow) represents the outcome of interest (e.g., a particular link-wise connectivity coefficient) and is longitudinally measured by 10 measures T1 to T10 (blue rectangles). A second cross-sectional measurement, Tx (green rectangle), augments the estimation of the between-person differences. The respective data set is organized with a mutually missing data scheme, that is, all cells of the longitudinal measurements are missing for cross-sectional data rows and vice versa.

DOI: https://doi.org/10.7554/eLife.35718.009

is outweighed by noise by a factor of about 4). The average ICC in our analysis is 0.28 and, thus, very much in line with the original analysis. Compared to *Pannunzi et al., 2018*, we find a compressed range of ICCs from 0.0 to 0.55 and second the claim that rsFC obtained from 5 min scans performs poorly as a marker for individual subjects (also see *Gordon et al., 2017*).

## Discussion

### When the true scores are changing: Extending ICED to growth curve modeling

So far, we have assumed that the construct of interest does not change over time. Thus, any change between repeated measures was assumed due to unsystematic variability, that is, noise. But what if the construct of interest varies over time? For example, had we modeled all fifty measurements from the day2day study that spanned roughly six month, we would have confounded reliability and lack of stability. it is very likely that the difference between repeated measures in the beginning and at the end of the study represent a mixture of measurement error and true within-person short-term variability, long-term change, or both (also see *Nesselroade, 1991*). When assessing reliability over repeated measures in practice, one seeks avoiding this problem by reducing the interval between measurements. At the same time, one is interested in independent measurements, and the degree of dependence may increase with shorter time spans as the chance of item-specific or construct-general temporal effects that may affect multiple measurements may artificially increase the reliability estimate. If, however, measurements are numerous or if the reliability estimate must be obtained from an existing study with a considerable time lag between measurements, it is likely that true change in the construct is present, and that persons differ regarding its magnitude, direction, or both. If substantive change is not accounted for, reliability estimates are biased towards lower values (*Brandmaier et al., 2018*). The resulting biased measure may still be useful when interpreted as a stability coefficient, while keeping in mind that instability may be caused by change as well as imprecise measurement. What is, however, the best strategy when we wish to know whether true scores have changed?

Elsewhere, we have applied the logic presented here to linear latent growth curve models (*Brandmaier et al., 2015*; *Brandmaier et al., 2018*; *von Oertzen and Brandmaier, 2013*). Effective

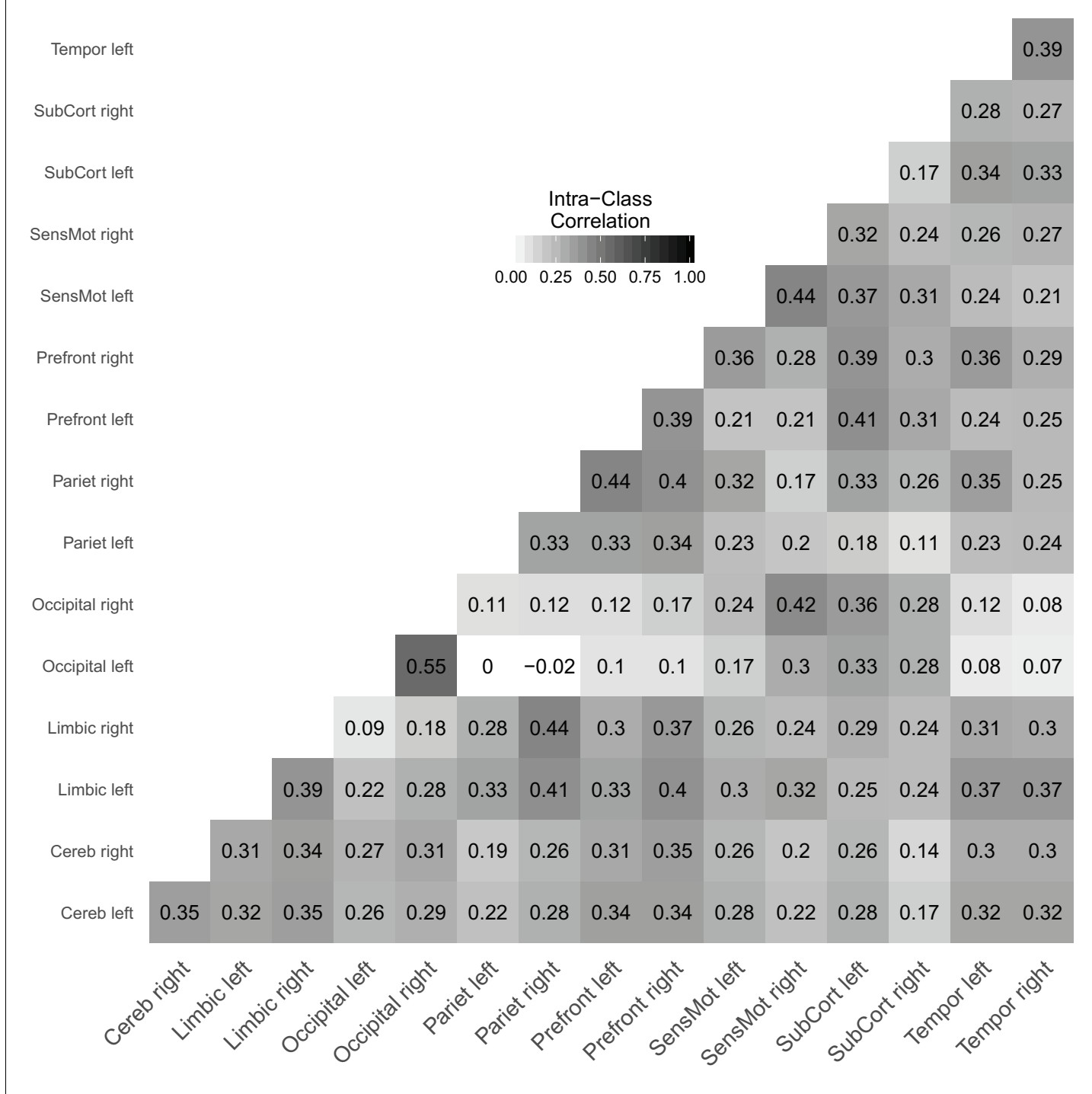

**Figure 7.** Link-wise reliability based on combined cross-sectional and longitudinal samples from the day2day study.
DOI: https://doi.org/10.7554/eLife.35718.010

error of the change component (or, slope) in a latent growth curve model reflects the precision with which a growth curve model can measure between-person differences in change. By scaling the magnitude of individual differences in change (i.e., between-person variance in slope) with effective error, we obtain effective curve reliability (ECR; *Brandmaier et al., 2015*). Major components of effective error for individual differences in change are the number of measurement occasions, the

temporal arrangement of measurement occasions, the total study time span, and instrument reliability. We have shown that effective error, reliability, and statistical power are all potentially useful measures that quantify the sensitivity of a longitudinal design, or any repeated measures design (for example, multiple sessions within a day), to measure individual differences in change (*Brandmaier et al., 2015*; *Brandmaier et al., 2018*). All these measures may be used for *a priori* design optimization. Such optimization entails either trading-off multiple design factors against each other, while keeping power constant, or changing power as a function of the various design factors and treating them as important measures to communicate reliability of change beyond cross-sectional reliability.

## Intra-class effect decomposition of group differences and interactions among error sources

In Section 4, we have discussed a research design that is optimal in factorizing the total error variance into three orthogonal error components of day-specific, session-specific, and unspecific residual variance. Optimality referred to a design that comprises the smallest number of measurements necessary to identify the sought-after error components. However, the ICED framework easily generalizes to more complex designs. For example, with a greater number of sessions, it would be possible to identify additional sources of error, such as experimenter-specific or site-specific errors. We can think of this framework as a variance decomposition approach just as in regular analysis of variance (see *Noble et al., 2017*), with the only difference that we are not interested in the sources of true score variance with the residuals set aside but rather in the decomposing the error score variance.

In the example reported in Section 4, we only examined main effects of day and session. Note, however, that the ICED method also can handle interaction effects. For example, we may be interested if there is an interaction of day and session, that is, if it matters on which day repositioning happened. To test this interaction, one can easily add a second group to the design presented earlier, with two sessions at Day one and one session at Day two (i.e., the mirror image of the current design, in which there is one session at Day one and two sessions at Day two). In this model, we could estimate the ICED components separately for each group. To test a potential interaction, we would state a null hypothesis of no differences in error variance across groups for the session effect. This is a null model of no interaction between session and order-of-day. Explicitly testing the initial model against the restricted null model yields a $\chi^2$ significance test of the interaction. Now, imposing this equality constraint on the session effect across groups would effectively test for the presence of reliable session by day interaction (e.g., does it make a difference whether repositioning within a day takes place at Day one or Day two). One could also conduct the same study with different groups, such as children, older adults, or patients with a particular disease or condition to evaluate group differences in day and session error contributions.

## Summary

In this paper, we have discussed the distinction and complementarity of ICC and CV in gauging reliability of brain imaging measures, a topic that thus far has received only limited attention. Considering the increasing demand for longitudinal and multi-center studies, there is a dire need for properly evaluating reliability and identifying components that contribute to measurement error. ICC and CV, as measures of (relative) precision, or reliability, fundamentally relate information about lasting properties of the participants to the precision with which we can measure this information over repeated assessments under the assumption of no change in the underlying construct. We have shown how the generality of the SEM approach (cf. *McArdle, 1994*) may be leveraged to identify components of error sources and estimate their magnitude in more complex designs in more comprehensive and general ways than achievable with standard ANOVA-based ICC decompositions. The underlying framework for deriving the individual error components as factors of reliability is closely related to Cronbach's generalizability theory (or G-Theory; *Cronbach et al., 1972*), which was recently expressed in a SEM framework (*Vispoel et al., 2018*). Our approach is similar to those approaches but was derived using the power equivalence logic (*von Oertzen, 2010*) to analytically derive effective error and reliability scores in a SEM context. This means that our approach easily generalizes to complex measurement designs beyond standard ANOVA, and that effective error, ICC, $ICC_2$ can

automatically be derived using *von Oertzen (2010)* algorithm from any study design rendered as a path diagram or in matrix-based SEM notation.

As noted at the beginning of our article, ICC and CV represent two perspectives on reliability that correspond to a fundamental divide of approaches to the understanding of human behavior: the experimental and the correlational (individual differences), each coming with its own notion of reliability (*Cronbach, 1957*; *Hedge et al., 2018*). In experimental settings, reliable effects are usually those that are observed on average, that is, assumed to exist in most individuals. To facilitate detection of such effects, the within-person variability must be low in relation to the average effect. The experimental approach is therefore compatible with the CV perspective. In individual difference approaches, reliable effects distinguish well between persons, which is only true if the within-person variability is low in relation to the between-person variability. The two notions of reliability are associated with competing goals; hence, it is not surprising that robust experimental effects often do not translate into reliable individual differences (*Hedge et al., 2018*).

In addition to ICC and CV, other reliability indices have been reported. When researchers compare the similarity of sets, as in gauging the overlap of voxels identified in two repeated analyses of the same subject, the Sørensen–Dice similarity coefficient (or, Dice coefficient; *Dice, 1945*; *Sørensen, 1948*) is often used. Since we are focusing on the reliability of derived continuous indices (e.g., total gray matter volume, fractional anisotropy or indices of myelin water fraction in a region of interest, or link-wise resting state functional connectivity), we did not consider the Dice coefficient here. Others have used the Pearson product moment correlation coefficient, $r$, to quantify the consistency of test scores across repeated assessments. The linear correlation is a poor choice for reliability assessment because due to its invariance to linear transformation, it is insensitive to mean changes (*Bartko, 1966*). Moreover, it is limited to two-occasion data. Therefore, we have also not considered Pearson's $r$ here.

## Outlook

Effective error variance partitioning as described above can be useful for communicating absolute precision of measurement, on its own and complimentarily with reliability. Importantly, one needs to specify what kind of reliability is being sought: reliability with respect to an anchoring point (e.g., the scale's zero) or with respect to the heterogeneity in the population. It needs to be emphasized that ICC can only be large if there are individual differences across persons in the measure of interest. Critics of ICC-based approaches to estimating reliability have argued that this method confounds group heterogeneity in the outcome of interest and measurement precision, and therefore must 'be perceived as an extremely misleading criterion for judging the measurement qualities of an instrument.' (*Willett, 1989*, p. 595). We strongly disagree with this narrow view of measurement quality. In the proverbial sense, 'one man's trash is another man's treasure,' and what some may view as a 'confound,' is for others a virtue of the measure in as much as it determines the capability of detecting heterogeneity in the population. However, the ICC may reveal nothing about the trial-to-trial differences expressed as deviations in the actual unit of measurement; those are better represented by the within-person standard deviation or standardized versions of it. We maintain that ICC is the appropriate measure of reliability when assessing diagnostic instruments and especially while focusing on individual differences.

In this article we introduced ICED as a variance-partitioning framework to quantify the contributions of various measurement context characteristics to unreliability. ICED allows researchers to (1) identify error components; (2) draw inferences about their statistical significance and effect size; and (3) inform the design of future studies.

Given the remarkable pace of progress in human brain imaging, researchers often will be interested in the (yet unknown) reliability of a new neuroimaging measure. Whether this reliability is sufficient can roughly be decided using thresholds, which essentially are a matter of consensus and conventions. For example, reliability larger than 0.9 is often regarded as excellent, as it implies a signal to noise ratio of 10:1. However, there may be good reasons to adopt less conservative thresholds (e.g., *Cicchetti and Sparrow, 1981*). In addition, using ICED, researchers can go beyond a summary index of ICC and instead report the magnitudes of individual variance components that contribute to lowering the overall ICC. These different components may differ in their methodological and practical implications. Often, researchers will be interested in using inferential statistics to test whether each of the individual variance components differs from zero and, maybe, whether the components

differ from each other. Finally, the results of these analyses can guide researchers in their subsequent attempts to improve measurement reliability. For instance, using ICED, researchers may discover that a hitherto overlooked but remediable source of error greatly contributes to unreliability, and work on improving the measurement properties influencing this component. Also, researchers may ask what combinations of measurements are needed to attain a target reliability (*Noble et al., 2017*) while optimizing an external criterion such as minimizing costs or participant burden (*Brandmaier et al., 2015*).

To conclude, we hope that the tools summarized under ICED will be applied in human brain imaging studies to index overall reliability, and to identify and quantify multi-source contributions to measurement error. We are confident that the use of ICED will help researcher to develop more reliable measures, which are a prerequisite for more valid studies.

## Acknowledgements

We thank Muzamil Arshad and Jeffrey A Stanley from the Department of Psychiatry and Behavioral Neuroscience, School of Medicine, Wayne State University, Detroit, Michigan, for providing the raw data on Myelin Water Fraction measurements. This work was supported by European Union's Horizon 2020 research and innovation programme under grant agreement No. 732592: 'Healthy minds from 0–100 years: optimizing the use of European brain imaging cohorts ('Lifebrain')' to AB, SK and UL, and by NIH grant R01-AG011230 to NR.

## Additional information

### Funding

| Funder | Grant reference number | Author |
|---|---|---|
| Horizon 2020 Framework Programme | 732592 | Andreas M Brandmaier<br>Simone Kühn<br>Ulman Lindenberger |
| Max-Planck-Gesellschaft | | Andreas M Brandmaier<br>Elisabeth Wenger<br>Nils C Bodammer<br>Naftali Raz<br>Ulman Lindenberger<br>Simone Kühn |
| National Institutes of Health | R01-AG011230 | Naftali Raz |

Open-access funding. The funders had no role in study design, data collection and interpretation, or the decision to submit the work for publication

### Author contributions

Andreas M Brandmaier, Conceptualization, Data curation, Formal analysis, Visualization, Methodology, Writing original draft, Writing—review and editing; Elisabeth Wenger, Nils C Bodammer, Conceptualization, Methodology, Writing—review and editing; Simone Kühn, Resources, Data curation, Investigation, Writing—review and editing; Naftali Raz, Conceptualization, Resources, Data curation, Supervision, Methodology, Writing—review and editing; Ulman Lindenberger, Conceptualization, Supervision, Methodology, Writing—review and editing

### Author ORCIDs

Andreas M Brandmaier http://orcid.org/0000-0001-8765-6982
Naftali Raz http://orcid.org/0000-0002-5080-2138
Ulman Lindenberger http://orcid.org/0000-0001-8428-6453

### Decision letter and Author response

Decision letter https://doi.org/10.7554/eLife.35718.016
Author response https://doi.org/10.7554/eLife.35718.017

## Additional files

### Data availability

The dataset on myelin water fraction measurements is freely available at https://osf.io/t68my/ and the link-wise resting state functional connectivity data are available at https://osf.io/8n24x/.

The following previously published datasets were used:

| Author(s) | Year | Dataset title | Dataset URL | Database, license, and accessibility information |
|-----------|------|---------------|-------------|-------------------------------------------------|
| Pannunzi M, Hindriks R, Bettinardi R, Wenger E, Lisofsky N, Mårtensson J, Butler O, Filevich E, Becker M, Lochstet M, Lindenberger U, Kühn S, Deco G | 2018 | Resting-state fMRI correlations: from link-wise unreliability to whole brain stability | https://osf.io/8n24x/ | Publicly available at Open Science Framework |
| Arshad M, Stanley J A, Raz N | 2018 | Reliability of Myelin Water Fraction in ALIC | https://osf.io/t68my/ | Publicly available at Open Science Framework |

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
