## [Decision Letter]

Thank you for submitting your article "Assessing Reliability in Neuroimaging Research Through Intra-Class Effect Decomposition (ICED)" for consideration by *eLife*. Your article has been reviewed by two peer reviewers, and the evaluation has been overseen by a Reviewing Editor and Sabine Kastner as the Senior Editor. The following individual involved in review of your submission has agreed to reveal his identity: Nico Dosenbach (Reviewer #3).

The reviewers have discussed the reviews with one another and the Reviewing Editor has drafted this decision to help you prepare a revised submission.

Summary:

In this article Brandmaier et al. discuss the differences between coefficient of variation (CV) and intra-class correlation coefficient (ICC) and then introduce a novel measure called the intra-class effect decomposition (ICED). The ICED estimates sources of error by explicitly modeling the contributions of each latent source of variance (e.g., person, session, day) to each scan via confirmatory SEM. These sources are then combined into a power equivalence form of the ICC, and used to estimate ICC under different conditions. The analyses of Arshad et al. (2017) are replicated using this method in order to demonstrate that different forms of the ICC (using different error components) can be estimated from the full dataset.

Essential revisions:

1) The motivation for the approach, and what it adds over existing methods, needs to be clarified. The authors state that the "key feature of this method is its ability to distinguish among multiple sources of unreliability, with understanding that not all errors are equally important and meaningful in repeated-measures design", and again highlight this in the subsection “An Empirical Example: Myelin Water Fraction Data from Arshad et al. (2017)”, last paragraph. However, these benefits are already present in ANOVA-based ICC. Generalizability Theory (G-Theory; Webb and Shavelson, 2005) has been used in neuroimaging to decompose error into constituent sources and re-estimate ICCs (e.g., Gee et al., 2015, Noble et al., 2017). Despite this crucial point, the authors only mention G-Theory in the third paragraph of the subsection “When the true scores are changing: Extending ICED to growth curve modeling”. From a practical standpoint, the estimates of the ICC2 are very similar to the estimates of Arshad et al. (2017). Although they use a subset of the data to estimate back-to-back versus repositioned ICCs, this can be estimated with the full data and an ANOVA using G-Theory.

While the authors state another motivating virtue – the ability to use "likelihood ratio tests to efficiently assess whether individual variance components significantly differ from zero" – they do not acknowledge that this is also possible in a traditional ANOVA framework, e.g., via simple F-test.

The value of this approach, which should be at the heart of its motivation, is that it provides a theoretically more valid way of estimating error from multiple sources, particularly in complex and time-dependent designs. The ANOVA and repeated-measures ANOVA become increasingly invalid with the complexity of the design. In addition, the SEM framework allows the user to test different assumptions about the structure of the model (e.g., tau-equivalent vs. con-generic). The aforementioned subsection starts to hint at this, though this should be the core feature throughout. Note however, that this is a theoretical argument; it is difficult to demonstrate the "utility" of this method over the simpler ANOVA, especially for simpler designs where the gains may be small.

2) Subsection “Intra-class effect decomposition (ICED)”, fifth paragraph: Significantly, this form of the ICC, with error variance divided by k (here, 4), represents an average measure over multiple scans. Reliability of averages are not relevant to most purposes, where a single scan is of interest. Arshad et al. get similar values (ICC=0.83) as the value in the third paragraph of the subsection “An Empirical Example: Myelin Water Fraction Data from Arshad et al. (2017)” in ALIC with repositioning, though it is difficult to tell whether they are using average measures from their methods (the exact form of the ICC is not given).

3) The extensive discussion in the Introduction/subsection “Comparing CV and ICC: Different but compatible conceptions of signal and noise” of the relative pros and cons of ICC and CV, alongside language about reconciling disparate approaches (Abstract and Introduction, first paragraph), leads the reader to anticipate a measure that reconciles CV and ICC. However, this is not the case. Therefore, the in-depth discussion of CV therefore seems out of place. The authors also return to a confusing/imprecise discussion about this in the fourth paragraph of the subsection “When the true scores are changing: Extending ICED to growth curve modeling”.

4) Subsection “Intra-class effect decomposition (ICED)”, fourth paragraph: More details are needed for the SEM estimation procedure. For instance, does "identical and fixed" mean tau-equivalent same weight across paths, but that weight is freely estimated? Since each residual is estimated separately for each scan, how are these residuals then combined into a single residual error term for the ICC?

5) Subsection “Intra-class effect decomposition (ICED)”, fifth paragraph: Do the reliability curves mentioned here refer to explicitly varying the magnitude of the error terms? If so, what is the utility of this? Or does this refer to vary the number of measurements in a Decision Study, as in Noble et al. (2017)?

---

## [Author Response]

Essential revisions:1) The motivation for the approach, and what it adds over existing methods, needs to be clarified. The authors state that the "key feature of this method is its ability to distinguish among multiple sources of unreliability, with understanding that not all errors are equally important and meaningful in repeated-measures design", and again highlight this in the subsection “An Empirical Example: Myelin Water Fraction Data from Arshad et al. (2017)”, last paragraph. However, these benefits are already present in ANOVA-based ICC. Generalizability Theory (G-Theory; Webb and Shavelson, 2005) has been used in neuroimaging to decompose error into constituent sources and re-estimate ICCs (e.g., Gee et al., 2015, Noble et al., 2017). Despite this crucial point, the authors only mention G-Theory in the third paragraph of the subsection “When the true scores are changing: Extending ICED to growth curve modeling”.

We are grateful for this comment. We were not aware of the fact that previous work had applied G-Theory to neuroimaging data and we are glad that this was brought to our attention. We now acknowledge these previous applications at various places throughout the manuscript. Furthermore, we draw a stronger link to G-Theory much earlier in the manuscript and also throughout the manuscript. For example, in the Introduction:

“Below, we focus on individual differences as we present a general and versatile method for estimating the relative contributions of different sources of error in repeated-measures designs. […] In this sense, it is tightly linked to G-Theory, which has been used successfully before in assessing reliability of neuroimaging measures (Gee et al., 2015; Noble et al., 2017).”

Or, in the Discussion:

“The underlying framework for deriving the individual error components as factors of reliability is closely related to Cronbach’s generalizability theory (or G-Theory; Cronbach et al., 1972), which was recently expressed in a SEM framework (Vispoel, Morris, and Kilinc, 2017). […] This means that our approach easily generalizes to complex measurement designs beyond standard ANOVA, and that effective error, ICC, ICC_2_ can automatically be derived using von Oertzen’s (2010) algorithm from any study design rendered as a path diagram or in matrix-based SEM notation.”

From a practical standpoint, the estimates of the ICC2 are very similar to the estimates of Arshad et al. (2017). Although they use a subset of the data to estimate back-to-back versus repositioned ICCs, this can be estimated with the full data and an ANOVA using G-Theory.While the authors state another motivating virtue – the ability to use "likelihood ratio tests to efficiently assess whether individual variance components significantly differ from zero" – they do not acknowledge that this is also possible in a traditional ANOVA framework, e.g., via simple F-test.

This observation is correct. We now state that our results can equivalently be derived using ANOVA-based ICCs following G-Theory logic and we acknowledge that the F-test is applicable in the first example using Arshad’s data. We further motivate our approach by stating that we are able to test more complex hypotheses using our framework, involving hierarchically nested structures, multi-group models, or informative hypothesis testing. For example:

“The key feature of the method, however, is its ability to distinguish among multiple sources of un-reliability, with the understanding that not all sources of error and their separation are important and meaningful in repeated-measures designs. […] The ability to adequately model these relationships and visually represent them in path diagrams is a virtue of our approach.”

and

“Beyond, the generality of SEM allows us to test complex hypotheses involving hierarchically nested error structures or multi-group models and allows inference under missing data or by evaluating informative hypotheses (de Schoot, Hoijtink, and Jan-Willem, 2011) whereas ANOVA-based approaches increasingly become invalid with the complexity of the design.”

The value of this approach, which should be at the heart of its motivation, is that it provides a theoretically more valid way of estimating error from multiple sources, particularly in complex and time-dependent designs. The ANOVA and repeated-measures ANOVA become increasingly invalid with the complexity of the design. In addition, the SEM framework allows the user to test different assumptions about the structure of the model (e.g., tau-equivalent vs. con-generic). The aforementioned subsection starts to hint at this, though this should be the core feature throughout. Note however, that this is a theoretical argument; it is difficult to demonstrate the "utility" of this method over the simpler ANOVA, especially for simpler designs where the gains may be small.

We are thankful for this comment. We have added specific references to the ANOVA framework and have made clearer when the SEM framework is preferable. We have added a complete new analysis to illustrate a complex design in which we benefit from the flexibility that the SEM framework has to offer. Please see the newly added text describing our reliability analysis of the data from the day2day study.

Concerning the assumptions about the measurement model, we clarified that we are assuming a parallel measurement model. We acknowledge that a specific virtue of our framework is the generality to allow for modeling and testing other forms of measurement models (con-generic, tau-equivalent) but we believe that most use-cases as well as our demonstrations will assume a parallel model. We have revised the whole section introducing the model. Here is a revised excerpt from the manuscript:

“In classical test theory, this is referred to as a parallel model, in which the construct is measured on the same scale with identical precision at each occasion. […] Note that, in the SEM framework, we also can extend the parallel model to more complex types of measurement models (e.g., con-generic or tau-equivalent models) that allow for different residual error variances or different factor loadings.

2) Subsection “Intra-class effect decomposition (ICED)”, fifth paragraph: Significantly, this form of the ICC, with error variance divided by k (here, 4), represents an average measure over multiple scans. Reliability of averages are not relevant to most purposes, where a single scan is of interest. Arshad et al. get similar values (ICC=0.83) as the value in the third paragraph of the subsection “An Empirical Example: Myelin Water Fraction Data from Arshad et al. (2017)” in ALIC with repositioning, though it is difficult to tell whether they are using average measures from their methods (the exact form of the ICC is not given).

It is correct to note that the ICC2 relates to the reliability of an average obtained from multiple scans. However, we disagree that ICC2 is not important. In fact, it is the primary value of interest when our framework is used to inform a future study design, in which one can decide how many scans will be performed and under what conditions the measurements will take place (e.g., two runs in a single session or two runs in two sessions, etc.). For example, Noble et al. (2017) use a ICC2-type measure to investigate the effects of total scan time and number of sessions on (average) reliability. We have strengthened this point in response to another comment below. Newly added excerpts to the manuscript are:

“In many cognitive neuroscience studies, one may be interested in construct-level reliability, and not only in reliability of indicators (i.e., observed variables). […] This construct reliability is captured by ICC_2_ (Bliese, 2000).”

and

“In sum, ICC is a coefficient describing test-retest reliability of a measure (this was also referred to as short-term reliability or intra-session reliability by Noble, 2017) whereas ICC_2_ is a coefficient describing test-retest reliability of an underlying construct (an average score in parallel models) in a repeated-measures design (this was also referred to as long-term reliability or intersession reliability by Noble, 2017).”

We have also clarified the connection between Arshad et al.’s original results and our results. Revised excerpt from the manuscript:

“Arshad et al. (2017) only reported pair-wise ICCs, based either only on the two back-to-back sessions of a single day, or on a single session of each day (again omitting a third of the available data). […] Similarly, we can derive the reliability of a single measurement, had we measured only the two back-to-back sessions, achieving the identical result:

ICC=0.94”.

3) The extensive discussion in the Introduction/subsection “Comparing CV and ICC: Different but compatible conceptions of signal and noise” of the relative pros and cons of ICC and CV, alongside language about reconciling disparate approaches (Abstract and Introduction, first paragraph), leads the reader to anticipate a measure that reconciles CV and ICC. However, this is not the case. Therefore, the in-depth discussion of CV therefore seems out of place. The authors also return to a confusing/imprecise discussion about this in the fourth paragraph of the subsection “When the true scores are changing: Extending ICED to growth curve modeling”.

We thank you for this comment. From our discussions with various researchers both within our own group and with others, it became clear that there is some confusion about when (or why at all) one should use CV or ICC to assess reliability. We believe such a thorough discussion is much needed and indispensable before one can fully acknowledge the importance of our framework.

Still, we agree that we may have built up some wrong expectations in the Abstract and the Introduction, so we have changed the manuscript in several places to clarify that we are addressing two related goals, here. First, we motivate the use of ICC while acknowledging its relation to CV, and, second, we introduce a general variance- partitioning framework related to G-theory that allows us to better understand sources of unreliability in our neuroimaging measurements.

We have clarified the Discussion in the following way:

“As noted at the beginning of our article, ICC and CV represent two perspectives on reliability that correspond to a fundamental divide of approaches to the understanding of human behavior: the experimental and the correlational (individual differences), each coming with its own notion of reliability (Cronbach, 1957; Hedge, Powell, & Sumner, 2017). […] The two notions of reliability are associated with competing goals; hence, it is not surprising that robust experimental effects often do not translate into reliable individual differences (Hedge et al., 2017).

4) Subsection “Intra-class effect decomposition (ICED)”, fourth paragraph: More details are needed for the SEM estimation procedure. For instance, does "identical and fixed" mean tau-equivalent same weight across paths, but that weight is freely estimated? Since each residual is estimated separately for each scan, how are these residuals then combined into a single residual error term for the ICC?

We thank you for this comment. We have clarified details of the estimation procedure (see our response to item 1, for example, regarding the parallel model and potential extensions to tau-equivalent models). In fact, residuals are not necessarily separately estimated for each scan. The residual errors are sums of the individual variance components and their composition depends on the path diagram which again represents a given study design. They can be directly read off from the model-implied covariance matrix and the actual residual error components can be either be computed or manually derived using path-tracing rules. We have clarified the entire related section and added, for example:

“The model-implied covariance matrix has the total variances for each observed variable in the diagonal. It can be analytically or numerically derived using matrix algebra (McArdle, 1980) or path-tracing rules (Boker, McArdle, and Neale, 2002), and is typically available in SEM computer programs (e.g., von Oertzen, Brandmaier, and Tsang, 2015).”

We also added further clarifications on how to compute the residual error (“effective error”) for *ICC_2_*, for example:

“The effective error is a function of all error components and the specific composition depends on a specific study design. […] Effective error can be computed using the algorithm provided by von Oertzen (2010) and for some models, analytic expressions are available (see the multi-indicator theorem in von Oertzen, Hertzog, Lindenberger, and Ghisletta, 2010).”

5) Subsection “Intra-class effect decomposition (ICED)”, fifth paragraph: Do the reliability curves mentioned here refer to explicitly varying the magnitude of the error terms? If so, what is the utility of this? Or does this refer to vary the number of measurements in a Decision Study, as in Noble et al. (2017)?

Correct. We have made explicit the link to Noble’s very convincing application of this idea. We also explicitly link to the idea of decision studies in general in a new paragraph in the Discussion. Here is an excerpt from the revised manuscript:

“For our hypothesized measurement model that includes multiple measurements and multiple variance sources, the analytic solution of *ICC_2_* allows us, for instance, to analytically trace reliability curves depending on properties of a design, such as the number of sessions, number of runs per sessions, number of sessions per day, or varying magnitudes of the error component. Of note, this corresponds to a D-study in G-Theory that can demonstrate, for example, how total session duration and number of sessions influence resting state functional connectivity reliability (see Noble et al., 2017).”